# Liquid Crystal-Based Organosilicone Elastomers with Supreme Mechanical Adaptability

**DOI:** 10.3390/polym14040789

**Published:** 2022-02-18

**Authors:** Zhe Liu, Yuqi Xiong, Jinghao Hao, Hao Zhang, Xiao Cheng, Hua Wang, Wei Chen, Chuanjian Zhou

**Affiliations:** 1School of Materials Science and Engineering, Shandong University, Jinan 250061, China; 201820362@mail.sdu.edu.cn (Z.L.); 201784000014@sdu.edu.cn (J.H.); 202020494@mail.sdu.edu.cn (H.Z.); chengxiao@sdu.edu.cn (X.C.); hwang@sdu.edu.cn (H.W.); 2CAS Key Laboratory of Soft Matter Chemistry, University of Science and Technology of China, Hefei 230026, China; azong2021@mail.ustc.edu.cn

**Keywords:** liquid crystal-based organosilicon elastomers, supreme mechanical adaptability, thiol-ene “click” reaction

## Abstract

Elastomers with supreme mechanical adaptability where the increasing stress under continuous deformation is significantly inhibited within a large deformation zone, are highly desired in many areas, such as artificial muscles, flexible and wearable electronics, and soft artificial-intelligence robots. Such system comprises the advantages of recoverable elasticity and internal compensation to external mechanical work. To obtain elastomer with supreme mechanical adaptability, a novel liquid crystal-based organosilicon elastomer (LCMQ) is developed in this work, which takes the advantages of reversible strain-induced phase transition of liquid crystal units in polymer matrix and the recoverable nano-sized fillers. The former is responsible for the inhibition of stress increasing during deformation, where the external work is mostly compensated by internal phase transition, and the latter provides tunable and sufficient high tensile strength. Such LCMQs were synthesized with 4-methoxyphenyl 4-(but-3-en-1-yloxy)benzoate (MBB) grafted thiol silicone oil (crosslinker-*g*-MBB) as crosslinking agent, vinyl terminated polydimethylsiloxane as base adhesive, and fumed silica as reinforcing filler by two-step thiol-ene “click” reaction. The obtained tensile strength and the elongation at break are better than previously reported values. Moreover, the resulting liquid crystal elastomers exhibit different mechanical behavior from conventional silicone rubbers. When the liquid crystal content increases from 1% (*w*/*w*) to 4% (*w*/*w*), the stress plateau for mechanical adaptability becomes clearer. Moreover, the liquid crystal elastomer has no obvious deformation from 25 °C to 120 °C and is expected to be used in industrial applications. It also provides a new template for the modification of organosilicon elastomers.

## 1. Introduction

Elastomers with supreme mechanical adaptability have been used and developed for several years [1,2,3], which have not only recoverable large deformation capacity, but also ability to compensate external work during continuous deformation [4,5]. For instance, for widely used connection devices, i.e., rubber ring, it is desirable to not only provide certain tensile strength for load-bearing but also show a stress plateau as along as possible for severe service condition. Numerous strategies have been adopted to achieve this goal [6,7]. Among them, the liquid crystal elastomer, as a kind of material which can dissipate energy through reversible internal phase transition under external stimulation and have the recoverable large deformation capacity [8,9,10], becomes one of the best choices for mechanical adaptability materials [11,12]. Since Küpfer and Finkelmann developed a method for the preparation of monodomain liquid crystal elastomers in 1991 [13], the research on liquid crystal elastomers has become a hot topic [14,15,16]. Liquid crystal elastomer, as a kind of functional soft material which can produce large volume contraction by light or heat stimulation [17,18,19], is mainly used to fabricate soft actuators or robots [20,21,22,23,24,25]. Yang et al. recently prepared photoresponsive gold nanorod/liquid crystal elastomers which can be repeatedly programmed to deform [26]. Additionally, Yakacki et al. used digital light processing technology to realize 3D printing of liquid crystal elastomers for rapid prototyping of ultra-light three-dimensional energy-absorbing structures [27]. At present, most liquid crystal elastomers are preoriented before crosslinking to obtain monodomain liquid crystal elastomers with better orientation to improve their deformation ability. However, reports related to the introduction of the polydomain liquid crystal phase have been rarely reported.

There are two basic criteria for elastomers with supreme mechanical adaptability, namely, the sufficiently high tensile strength at the onset of stress plateau and the length of the stress plateau. Most research on liquid crystal elastomers focuses on developing environmental response materials, such as temperature, light, and magnetic responsiveness [28,29,30,31]. And the mechanical tensile strength in previous literatures almost never exceeds 1 MPa [7,17,19,30,31]. Few research groups have systematically studied its mechanical adaptability due to the poor mechanical strength and synthetic challenge [32].

Here we developed a new method for preparing liquid crystal-based organosilicon elastomers with excellent mechanical adaptability and strength. We firstly synthesized liquid crystal-based organosilicon elastomers with crosslinker-*g*-MBB as crosslinking agent, vinyl terminated polydimethylsiloxane as base adhesive, and fumed silica as reinforcing filler by two thiol-ene “click” reaction strategies. Compared to the traditional liquid crystal elastomers synthesized by hydrosilylation reaction using Pt complex catalysts, our technology has three important advantages: (1) it avoids costly metal platinum catalyst and the reaction conditions are milder and more convenient [33,34]; (2) the hydrosilylation reaction usually results in a mixture of Markovnikov and anti-Markovnikov addition products, which complicate the characterization of the product [35], but thiol-ene “click” reaction is difficult to produce markovnikov addition product [36]; (3) compared to the previous one-step strategy [37,38], we can improve the mechanical strength of the elastomers to more than 4 MPa by using the reinforcing filler. The liquid crystal-based organosilicon elastomers with supreme mechanical adaptability fabricated by this method are expected to be used in industrial applications and it provides a new template for the modification of organosilicon elastomers.

## 2. Experimental Section

### 2.1. Materials

The 3-Mercaptopropylmethyldimethoxysilane and Hexamethyldisiloxane (MM) were purchased from Chenguang Chemical Co., Ltd. (Qufu, China). α,ω-Vinyl terminated polydimethylsiloxane (10,000 mPa.s) was purchased from Dayi Chemical Co., Ltd. (Laiyang, China). The 4-Methoxyphenyl 4-(3-Butenyloxy) benzoate (MBB) was bought from Titan Scientific Co., Ltd. (Shanghai, China). Trifluoromethanesulfonic acid was obtained from J&K Scientific. The 2,2-Dimethoxy-2-phenylacetophenone (DMPA) was obtained from Aladdin Chemical Corporation. Toluene (≥99.5%, Sinopharm Chemical Reagent Co., Ltd., Shanghai, China), dichloromethane (DCM, ≥99.5%, Sinopharm Chemical Reagent Co., Ltd., Shanghai, China), sulfuric acid (≥98%, Sinopharm Chemical Reagent Co., Ltd., Shanghai, China). The Nano-silica (H2000) was purchased from Wacker Chemie AG Corporation. Other reagents were used as received without further purification.

### 2.2. Characterization of Silicone Oils and Elastomers

Nuclear magnetic resonance spectroscopy (^1^H NMR) were recorded on a Bruker Avance 500 MHz spectrometer without tetramethylsilane as the internal standard and deuterated chloroform (CDCl_3_) was used as the solvent. Ubbelodhe viscometer (IVS-300, zonmon technology Co., Ltd., Hangzhou, China) was used to determine the viscosity-average molecular weight (*M*_v_) and silicone oil was dissolved in chlorobenzene to obtain a solution with a concentration of approximate 4 mg/mL. The apparent viscosity and rheological behavior were obtained on a rheometer (Brookfield R/S plus, AMETEK Brookfield, Middleborough, MA, USA). The samples were measured by Fourier Transform Infrared Spectroscopy (FTIR) (Tensor37, Bruker Company, Berlin, Germany) in the range of 400–4000 cm^−1^ and samples were pressed into potassium bromide (KBr) pellet prior to the measurements. The cross-linking density was obtained by the instrument (VTMR20-010V-T, Suzhou Niumag Analytical Instrument Corporation, Shanghai, China). XLD_2_ fitting method was selected for results. The test temperature was set at 35 °C. The differential scanning calorimetry (DSC) instrument (DSC8000, Perkin Elmer, Waltham, MA, USA) was used to evaluate the glass transition temperature, phase-transition temperature at a heating rate of 10 °C/min in the atmosphere of nitrogen. Thermogravimetric analysis (TGA) was determined with a Labsys Evolution TGA/DSC Synchronous Thermal Analyzer (Seteram, Lyon, France) at a heating rate of 10 °C/min. The test temperature ranged from 30 °C to 800 °C in air. The tensile tests of silicone elastomers were characterized by a universal tensile tester (Sansi Vertical Technology Co., Ltd., Shenzhen, China). The 45 °C tensile tests of liquid crystal-based organosilicon elastomers were tested by a universal tensile tester (Gotech Testing Machines Inc., Taiwan, China). The liquid crystal-based organosilicon elastomers were tested for tensile strength and elongation at break according to the GB/T528-2009 standard with a loading speed of 500 mm/min and the gauge length of 20 mm. The Shore A hardness was measured by the GT-GS-MB rubber hardness apparatus (Gotech Testing Machines Co., Ltd., Dongguan, China). The dynamic mechanical thermal analysis (DMA) were acquired in a tension mode with a DMA/DMTA (Electroforce 3200, Bose, USA). The specimen type was a rectangle with a width of 9 mm, a length of 20 mm, and a thickness of 2 mm. The testing temperature was ranged from −80 °C to 50 °C at a rate of 6 °C per minute and a constant frequency of 1 Hz. The liquid crystalline textures of crosslinker-*g*-MBB were performed on the Carl Zeiss-Axio Scope.A1 polarized optical microscope (POM) at a heating rate of 10 °C/min. The transmission electron microscope (TEM) images were recorded on a JNM-2100 instrument operating at an acceleration voltage of 20–200 kV. Point resolution: 0.194 nm, linear resolution: 0.14 nm. Cryo-ultramicrotomy was used to prepare specimens for TEM observation. Small angle X-ray scattering (SAXS) experiments were performed with the instrument (SAXSess mc2, Anton Paar, Graz, Austria) equipped with Kratky block-collimation system and a temperature control unit (TCS300, Anton Paar, Graz, Austria). The maximum 2θ Angle measured is 40°. Measuring range: 0.2~150 nm.

### 2.3. Synthesis of Crosslinker

Typically, sulfuric acid (200 mL, 5%) and 3-Mercaptopropylmethyldimethoxysilane (100 mL) were added in a 500 mL, three-necked, round-bottomed flask with a mechanical stirrer and placed in an oil bath thermostated at 90 °C. The mixture was kept reflux for 6 h to fully hydrolyze. The acid was removed with deionized water by a separatory funnel and the crude product was dried for 24 h. Then the mercaptopropyl siloxane prepolymer was obtained by filtration. Siloxane prepolymer (15 g), MM (1.81 g), and trifluoromethanesulfonic acid (10 ul) were added in a 50 mL, three-necked, round-bottomed flask with a mechanical stirrer. The reaction was carried out at 60 °C for 6 h. Trifluoromethanesulfonic acid was removed after washed three times with deionized water. Polymethylmercaptpropylsiloxane was harvested by vacuum distillation at 150 °C. Yield: 80%. ^1^H NMR (CDCL_3,_ 400 MHz): δ 0.07 (s, -Si(CH_3_)_3_), 0.60 (m, -SiCH_2_CH_2_-), 1.33 (m, -SH), 1.62 (m, -SiCH_2_CH_2_-).

### 2.4. Synthesis of Crosslinker-g-MBB

Polymethylmercaptpropylsiloxane, MBB, DMPA (0.1% *w*/*w*), and 20 mL toluene were added in a 50 mL round-bottomed flask with a mechanical stirrer and degassed by bubbling dry nitrogen gas for 1 h. The bottle was sealed and illuminated under 365 nm UV light for 20 min. The toluene was removed under reduced pressure to obtain product. Yield: 98%. ^1^H NMR (CDCL_3,_ 400 MHz): δ 0.10 (s, -Si(CH_3_)_3,_ -SiCH_3_CH_2_-), 0.64 (m, -SiCH_2_CH_2_-), 1.36 (m, -SH), 1.60–2.00 (s, -SiCH_2_CH_2_CH_2_-,-SCH_2_CH_2_CH_2_), 2.48–2.77 (m, -CH_2_CH_2_S-, -CH_2_CH_2_SH), 3.81 (s, -OCH_3_), 4.05 (m, -CH_2_CH_2_O), 6.94, 7.11, 8.12 (d, -OC_6_H_4_COOC_6_H_4_OCH_3_).

### 2.5. Fabrication of LCMQs

In a typical procedure, crosslinker-*g*-MBB was dissolved in DCM and transferred to a box. After the dichloromethane evaporated completely, α,ω-vinyl terminated polydimethylsiloxanes, 0.1% DMPA as the photoinitiator and fumed silica H2000 were added into the above box. Then mixing was carried out 5 times in a three-dimensional high-speed mixer for 40 s every time. Finally, mixture was placed onto the mold, the air eliminated, and illuminated by 365 nm UV light for 1 h. The milky white film was peeled off from the mold.

## 3. Results and Discussion

### 3.1. Synthesis and Characterization of Crosslinker, Crosslinker-g-MBB and LCMQs

The crosslinker-*g*-MBB was prepared in two steps: (i) synthesis of thiol silicone oil (crosslinker), (ii) thiol-ene click reaction of crosslinker and MBB. The synthesis of crosslinker and crosslinker-*g*-MBB was illustrated in Figure 1.

The degree of polymerization (DP) of –SH was determined to be ~32. The DP of –SH = 18/(I_5_-3*I_1_), where I_5_/I_1_ is the ratio of integrated intensity of peak 5 to that of peak 1 in the ^1^H NMR spectrum of Figure 1a. The number of grafting MBB was calculated to be approximately 15 based on the following equation: The number of grafting MBB = 32*(2*I_5_/I_2_), I_5_ and I_2_ is the ratio of integrated intensity of peak 5 to that of peak 2 in the ^1^H NMR spectrum of Figure 1b.

The grafting efficiency of MBB was almost 100%. In the FT-IR, the absorption peaks near 1260 cm^−1^, 784 cm^−1^ are assigned to Si-Me. The strong and wide absorption peaks near 1080 cm^−1^, 1025 cm^−1^ are attributed to Si-O-Si. The stretching vibration absorption peak of thiol group is near 2570 cm^−1^. Successful thiol-ene click reaction was also confirmed by FT-IR measurements due to the appearance of Ar-H (~3050 cm^−1^) and the ester group (~1730 cm^−1^) (Appendix A). In order to investigate the effect of cross-linking density on liquid crystal-based organosilicon elastomers, three kinds of silicone oils with different vinyl content (Appendix A) were prepared by equilibrium polymerization. The composition and characterization of silicone oils are listed in the Appendix A. The LCMQs were conveniently prepared by thiol-ene click reaction of crosslinker-*g*-MBB and α,ω-vinyl terminated polydimethylsiloxane. H2000 is used as reinforcing filler to improve the mechanical strength of elastomers. The details of all the synthesized liquid crystal-based organosilicon elastomers are summarized in Table 1.

### 3.2. Characterization of POM, SAXS and Rheological Determination

The POM measurements were firstly conducted for crosslinker-*g*-MBB to explore the liquid crystal phase. Figure 2a verifies the existences of nematic phases in the crosslinker-*g*-MBB. The liquid crystal phase disappears at 43 °C under POM observations (Figure 2b). The SAXS results indicate crosslinker-*g*-MBB shows a nematic scattering peak at room temperature. The crosslinker-*g*-MBB changes from the nematic state to the isotropic state at 43 °C and the scattering peak disappears, which is also completely consistent with the liquid crystal phase transition temperature measured by the POM (Figure 2c). Besides, rheological determination of crosslinker-*g*-MBB also confirmed the phase transition. The G′ and G″ of crosslinker-*g*-MBB decrease significantly when the temperature is around 43 °C and the damping factor increases from 0.3 to 2.2 at about 43 °C (Figure 2d,e). The results indicate that the crosslinker-*g*-MBB is in nematic state at room temperature, and the nematic to isotropic phase transformation temperature (*T*_NI_) of the crosslinker-*g*-MBB is 43 °C.

### 3.3. Determination of Crosslinking Density

Magnetic Resonance Crosslink Density Spectrometer was used to determine the crosslinking density of elastomers. Because the crosslinking density has a great relationship with the mechanical properties of elastomers, we prepared four kinds of samples with the same crosslinking density to explore the influence of liquid crystal content on the mechanical properties of elastomers, namely MQ 1 and LCMQs 2–4. It can be seen from Table 2 that the proportion of dangling chains of MQ 1 and LCMQs 2–4 are roughly equivalent, but the proportion of crosslinking chain of MQ 1 is obviously higher than that of LCMQs 2–4. This may be attributed to MQ 1 having no liquid crystal elements and not being affected by steric hindrance of liquid crystal elements, so the crosslinking is easier than that of LCMQs 2–4. The cross-linking density of LCMQ 4 to 6 gradually increases, which is caused by three kinds of silicone oils with different vinyl content used in the synthesis of LCMQ 4 to 6. With the reactivity increased, more sulfhydryl groups were reacted, so the proportion of dangling chains of LCMQ 4 was much higher than that of LCMQ 5 and 6. However, the proportion of crosslinking chains of LCMQ 6 is less than LCMQ 5, and this is because further reaction was inhibited by the quickly formed cross-linking structure of LCMQ6. In general, with the increase of crosslinking density, the mechanical strength increases. After the maximum value of crosslinking density, the mechanical strength will decrease greatly.

### 3.4. The Thermal Properties and the Thermal Stability Characterization of MQ 1 and the LCMQs

DSC was carried out to have a deep insight into the thermal properties of LCMQs. The glass transition temperature of dimethyl silicone rubber is near −123 °C and due to the flexible polydimethylsiloxane backbone, it could crystallize at low temperature [39,40]. The melting temperatures (*T*_m_) of MQ 1 and LCMQs remain almost invariant at −42 °C, where the introduction of liquid crystal elements has no significant effect on the *T*_m_ of the LCMQs. The *T*_m_ of LCMQs 4–6 slightly decrease with the increasing cross-linking density. The DSC curves of LCMQs 2–3 have only *T*_m_ during measurement, and liquid crystal phase transitions are not observed. This may be attributed to low fraction ratio of the liquid crystal content of LCMQ 2–3, which is beyond the detection sensitivity of DSC. The peaks of liquid crystal phase transition in LCMQs 4–6 appear at about 50 °C, which confirms that the crosslinking density have no influence on the temperature induced liquid crystal phase transition 9.

The TGA analysis was conducted to explore the influence of liquid crystal content on the thermal stability of the elastomers. Figure 3b shows that the thermal decomposition temperature of all the elastomers are all above 350 °C, so the LCMQs have excellent thermal stability. Moreover, the 5% weight loss temperature of MQ 1 to LCMQ 4 decreases from 425 °C to 389 °C, and the 10% weight loss temperature decreases from 450 °C to 433 °C (Appendix A). Therefore, with higher liquid crystal content, the elastomers show worse thermal stability. The liquid crystal element is mainly composed of carbon–carbon bonds, whose thermal stability is not as good as that of the siloxane linkage [41]. The excellent thermal stability of LCMQs guarantees the requirements of practical applications.

### 3.5. Mechanical Adaptability of MQ 1 and the LCMQs

Universal tensile tester was used to further investigate the influence of liquid crystal content and crosslinking density on the mechanical behavior of the elastomers, respectively. Actually, the liquid crystal-based organosilicon elastomers were not preoriented in the crosslinking process, so the samples obtained are polydomain liquid crystal elastomers [42]. *T*_NI_ is higher than room temperature, demonstrating that the samples are in the nematic phase during stretching. The stress-strain curves of MQ 1 and LCMQs 2–4 illustrate that the stress plateau area becomes more obvious with the increasing of liquid crystal content, which is similar to ‘soft elasticity’ (Figure 4a) [43], which can be attributed to the induced alignment of the liquid crystal elements along the direction of the applied stress during the stretching process [32,44]. Moreover, the tensile strength of LCMQs 2–4 decreased from 4.15 MPa to 2.16 MPa, and the elongation at break decrease from 677% to 601% (Table 3). This phenomenon does appear in the stress-strain curves of MQ 1 and LCMQs 2–4 at 45 °C, which further confirms our inference (Appendix A). In addition, the cross-linking density is also a crucial factor for this stress plateau area. With the increase of cross-linking density, the stress plateau area gradually disappears (Figure 4b). The increase of crosslinking density hinders the induced alignment of the liquid crystal elements along the direction of the applied stress during the stretching process. The tensile strength first increases and then decreases, and the elongation at break decreases from 601% to 242% (Table 3). Taking LCMQ 4 as an example, Figure 4d shows that the mechanical properties of elastomers do not change obviously at different tensile rates, which further proves the stability of mechanical properties of elastomers. This implies that the induced alignment of liquid crystal elements along the stress direction is very fast, and the change of tensile rate has little effect on its mechanical properties. The multiple cyclic stress-strain curves prove that the mechanical properties of the LCMQ 4 prepared by us are reliable and stable (Figure 4d). Although the liquid crystal elements are rearranged in the direction of the applied stress during the stretching process, the stress plateau region is reversible and the elastomers revert to its original polydomain state when the external stress disappears. Additionally, a comparison of the mechanical properties of liquid crystal-based organosilicon elastomers and other liquid crystal elastomers reported in references is shown in Figure 2f. The LCMQs prepared by us not only have supreme mechanical adaptability, but also excellent mechanical strength. It is clearly observed that both the tensile strength (up to 4 MPa) and the elongation (600%) in this work reached the top level of similar liquid crystal elastomers. Dumbbell samples with a thickness of 2 mm can bear nearly 1000 times their own weight, and strip samples can bear to 3500 times their own weight. Compared with the traditional thermal deformation liquid crystal elastomers, the LCMQs in this work have no obvious deformation from 25 °C to 120 °C (Appendix A), which is critical for industrial applications.

### 3.6. The Viscoelastic Properties of MQ 1 and the LCMQs

DMA was employed to evaluate the changes in the viscoelastic properties of liquid crystal-based organosilicon elastomers. According to the decreasing trend of storage modulus and loss modulus in Appendix A, it can be inferred that the *T*_m_ of the liquid crystal-based organosilicon elastomer is approximately −44 °C, which is in complete agreement with the results measured by DSC. The introduction of liquid crystal elements has a significant effect on the elastomers. When the temperature is lower than the *T*_m_, the storage modulus of MQ 1 is obviously larger than LCMQ 1–4, which implies that grafting liquid crystal elements affect the crystallization of the elastomers (Appendix A). When the temperature is higher than the *T*_m_, the crystallization of polydimethylsiloxane is destroyed, but the liquid crystal phase structure still exists. With more liquid crystal element, the more energy is consumed to convert the mechanical energy into internal energy, so the loss modulus will rapidly increase and then decrease (Appendix A). Besides the liquid crystal content, the crosslinking density should also be taken into account. With higher cross-linking density, it is more difficult for the segment to move (Appendix A). The introduction of liquid crystal elements improves the damping properties of the elastomers (Appendix A). Furthermore, increasing the cross-linking density will weaken this trend (Appendix A).

### 3.7. 2D USAXS Patterns of LCMQ 2, LCMQ 4 and LCMQ 6

In addition to the phase transition of liquid crystal content, the reorientation or redistribution of the filler structure is crucial for the mechanical behavior of silicone elastomer. To clarify the contribution of filler structure to the mechanical properties change, the time-resolved synchrotron radiation based USAXS technique was adopted. Here, the spatial detection range of USAXS used in current study covers 0.00475–0.24067 nm^−1^, which is sufficient to reflect the filler structure change during deformation. Figure 5a,b shows the 2D USAXS scattering patterns of LCMQs 2, 4, and 6 during stretching. With increasing strain, the initially isotropic scattering ring becomes dumbbell-shaped, which suggests the reorientation of filler structure. In order to deeply understand the strain-induced orientation behavior of filler content, the ‘mask’ operation was performed along the vertical and parallel directions to obtain the curve of the average scattering intensity *I(q)* versus *q* in the corresponding angle. The curves were fitted by Unified Guinnier/Power-law Approach to obtain the multiscale geometric information of particles:(1)$$I\left(q\right)=G\exp\left(-\frac{q^{2}R_{g}^{2}}{3}\right)+A\exp\left(-\frac{q^{2}R^{2}}{3}\right){\left\{{\left[\mathrm{erf}\left(\frac{qR_{g}}{\sqrt{6}}\right)\right]}^{3}/q\right\}}^{D_{m}}+B\exp\left(-\frac{q^{2}R^{2}}{3}\right)+C{\left\{{\left[\mathrm{erf}\left(\frac{qR_{g}}{\sqrt{6}}\right)\right]}^{3}/q\right\}}^{6-D_{s}}$$
where *G* is the Guinnier factor, *R_g_* and *R* are the radius of gyration of agglomerates and aggregates, respectively. *D_m_* is the mass dimension and *D_s_* is the surface dimension of particles. In a system without considering the surface roughness of particles, *D_s_* can be defined as 4. The fitting curves in vertical and parallel to the tensile direction are shown in Figure 5c while the information of *D_m_* is obtained by taking logarithm of axes before fitting.

In the following work, we compared the *R_g_*_⊥_ and *R_g_*_∥_ of the three samples when the strain is 200%, because the stress has an intriguing difference. As shown in Figure 5d, it is obvious that the presence of liquid crystal unit can affect the aggregation state of the fillers. LCMQ 4 has a large particle orientation scale, indicating that nematic liquid crystal can generate stress plateau area by promoting the aggregation of the filler under small deformation. However, the excessive cross-linking network in sample 6 increases the mechanical strength while making it difficult to withstand deformation inside the material, thus losing the adaptive function. The small value of *R_g_*_∥_ may indicate the rupture of the filler network structure. In short, the evolution of the complex network of fillers occurs at the millisecond level. As a multi-level structure analysis method, the time-resolved in-situ small-angle scattering method can provide us with enough reliable information about the evolution of structural statistical parameters. TEM observation reveals that H2000 is well-dispersed. Compared with MQ 1, LCMQ 4 has many equally distributed bubbles (Figure 5e,f). This invisible bubble structure contributes to the stress plateau area of elastomers. It may be obvious that the presence of these invisible bubbles affects the distribution of filler and promotes the aggregation of filler, which is also proved by the TEM results.

## 4. Conclusions

In summary, we reported a novel strategy towards liquid crystal-based organosilicon elastomers with excellent mechanical adaptability and systematically studied its strain-induced mechanical behavior. Both the tensile strength (up to 4 Mpa) and the elongation (600%) in this work reached the top level of liquid crystal elastomers. The stress plateau area becomes more obvious with the increasing liquid crystal content from 1% (*w*/*w*) to 4% (*w*/*w*). The increase of crosslinking density hinders the induced rearrangement of the liquid crystal elements along the direction of the applied stress during the stretching process. The introduction of liquid crystal phase improves the damping properties of the elastomers and increasing the cross-linking density can weaken this trend. Moreover, the LCMQs have no obvious deformation with the change of temperature. Although the introduction of liquid crystal phase causes the degradation of thermal stability of silicone elastomers, LCMQs still have good thermal stability. The presence of liquid crystal phase can affect the aggregation state of the fillers. Nematic liquid crystal can generate stress plateau area by promoting the aggregation of the filler under small deformation. We anticipate that the liquid crystal-based organosilicon elastomers fabricated by this method are expected to be used in industrial applications.

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
