# Peer review of "Liquid Crystal-Based Organosilicone Elastomers with Supreme Mechanical Adaptability"

_polymers, 2022, doi:10.3390/polym14040789_

Round 1

Reviewer 1 Report

In my opinion, the article is very interesting.

However, I have a few questions:

  1. Was DSC carried out in the heating-cooling-heating cycle? If not, why was only the first heating used?  Typically DSC carried out in the heating-cooling-heating is performed and data is collected from the first cooling ramp and the second heating ramp to exclude the samples’ thermal history. Please explain it.
  2. Please describe the FTIR spectra in detail and match the bands to the appropriate chemical groups.
  3. Why was the DTG [% / min] plot as a function of temperature not presented when presenting the TGA test results? Please present and discuss the DTG chart, as these are very important information.
  4. Shouldn't there be MPa instead of Mpa in the paragraph: "3.5 Mechanical adaptability of MQ 1and the LCMQs”"up to 4 Mpa" and "conclusion"
  5. Summary should be expanded

Reviewer 2 Report

The paper presents novel experimental results concerning elastomers with limited increasing stress under very large deformation (600% elongation). The approach involves the development of a novel polydomain side-chain liquid crystal-based organosilicone elastomer (LCMQ). The latter undergoes reversible strain-induced isotropic/nematic phase transition of liquid crystal units, thus limiting the stress increase (Fig.1b). The characterisation involves several techniques, including NMR, FTIR, dynamic mechanical thermal analysis, calorimetry, SAXS, TEM.... 

The paper is clearly written, with extensive coverage of the available literature on this interesting materials. The reported results account for a thorough characterisation effort. They  are convincingly presented and represent a significant contribution to the microscopic understanding of of liquid crystal elastomer.

Author Response

Thanks for the reviewer's postive comments.

Reviewer 3 Report

The authors have prepared and analysed novel LC-based organosilicone elastomers with notable mechanical adaptibility. The results are novel, the analysis is detailed and scientifically well presented. 
Therefore, I recommend the publication of the manuscript in the Polymers journal. 
Prior the publication, however, some minor modifications/corrections are needed. 

1) Grammatical or typing mistakes need to be corrected. Some examples:
"under continuously deformation" (Abstract); "previously literatures", "adapability" (Introduction); the use of singular vs. plural: "decrease" (end of p.6, last paragraph of Sect. 3.4); "the elastomer shows worse the thermal stability." (p.8); "phenomenon is not appeared" (p.9). 

2) Refs. [30,31] should become [28,29] according the text of the manuscript. 

3) the thickness of the elastomeric specimens (rectangular, Dumbell) should be given.

4) Scheme 1 should be presented first, before the Figure 1.

5) LCMQ 1 or LCMQ 2? Use the same notation in the Figures, Tables and text throughout the manuscript.

6) No Figure S3c mentioned in the text (p. 9).

7) Figures 4c and 4d are wrongly referenced in the text, as well as in the Figure caption. 

8) Subfigures (a) and (b) should be noted in the caption of Figure 4.

9) What are the dimensions of specimens shown in Figs. 4(a) and (b)?

10) Table 3 should reference Figure 4 (a) and (b), or Figure 5 (a) and (b)?

11) TEM image in Fig. 5(e) shows the edge of MQ 1, or the fillers aggregate heavily?

12) Figure 5(c) caption and in the text (p.12): instead of "in vertical and parallel to the tensile direction" 
a more precise formulation is: "perpendicular to, and parallel with the tensile direction".

13) page 12: NPs are not defined. It is better to use particles or fillers.

14) Ref. [39] has appeared in 1992; Ref. [44] is incomplete: vol.7, 23 (2022).

15) Figure caption of Fig. S4(a) is incomplete.
